# *Lactiplantibacillus plantarum* Postbiotics Suppress *Salmonella* Infection via Modulating Bacterial Pathogenicity, Autophagy and Inflammasome in Mice

**DOI:** 10.3390/ani13203215

**Published:** 2023-10-14

**Authors:** Aixin Hu, Wenxia Huang, Xin Shu, Shiyue Ma, Caimei Yang, Ruiqiang Zhang, Xiao Xiao, Yanping Wu

**Affiliations:** Key Laboratory of Applied Technology on Green-Eco-Healthy Animal Husbandry of Zhejiang Province, Zhejiang Provincial Engineering Laboratory for Animal Health Inspection & Internet Technology, Zhejiang International Science and Technology Cooperation Base for Veterinary Medicine and Health Management, China-Australia Joint Laboratory for Animal Health Big Data Analytics, College of Animal Science and Technology, College of Veterinary Medicine, Zhejiang Agriculture and Forestry University, Hangzhou 311300, China; axhu1026@163.com (A.H.); 18370687967@163.com (W.H.);

**Keywords:** probiotics, inactivated bacteria, metabolites, *Salmonella* Typhimurium, autophagy, inflammatory response

## Abstract

**Simple Summary:**

*Salmonella* infection is an urgent problem in animal husbandry, which causes salmonellosis in livestock and poses threats to human health through contaminated livestock products. As an alternative to antibiotics, probiotics play an important role in maintaining animal health. However, some probiotics, such as lactic acid bacteria, have limitations in storage and hostile environments. In this context, exploring the bacterial components or metabolites of probiotics has gradually drawn more research interest, and these are termed “postbiotics”. This study investigated the antibacterial effect of *Lactiplantibacillus plantarum* (LP) postbiotics and compared the effects to the live bacteria on intestinal health and autoimmunity in mice challenged with *Salmonella*. The results showed LP culture supernatant directly inhibited *Salmonella* growth and pathogenicity. LP postbiotics (the heat-killed bacteria and metabolites) showed similar or even superior effects to the active probiotic against *Salmonella* in mice. Furthermore, we found that LP postbiotics alleviated *Salmonella* infection via modulating bacterial pathogenicity, autophagy and inflammatory response. These results provide a theoretical basis for the protection of LP postbiotics against *Salmonella* and help to further explore its role in animal husbandry.

**Abstract:**

Our study aimed to explore the effects of postbiotics on protecting against *Salmonella* infection in mice and clarify the underlying mechanisms. Eighty 5-week-old C57BL/6 mice were gavaged daily with *Lactiplantibacillus plantarum* (LP)-derived postbiotics (heat-killed bacteria, LPB_inactive_; culture supernatant, LPC) or the active bacteria (LPB_active_), and gavaged with *Salmonella enterica* Typhimurium (ST). The Turbidimetry test and agar diffusion assay indicated that LPC directly inhibited *Salmonella* growth. Real-time PCR and biofilm inhibition assay showed that LPC had a strong ability in suppressing *Salmonella* pathogenicity by reducing virulence genes (*SopE*, *SopB*, *InvA*, *InvF*, *SipB*, *HilA*, *SipA* and *SopD*2), pili genes (*FilF*, *SefA*, *LpfA*, *FimF*), flagellum genes (*FlhD*, *FliC*, *FliD*) and biofilm formation. LP postbiotics were more effective than LP on attenuating ST-induced intestinal damage in mice, as indicated by increasing villus/crypt ratio and increasing the expression levels of tight junction proteins (Occludin and Claudin-1). Elisa assay showed that LP postbiotics significantly reduced ST-induced inflammation by regulating the levels of inflammatory cytokines (the increased IL-4 and IL-10 and the decreased TNF-α) in serum and ileum (*p* < 0.05). Furthermore, LP postbiotics inhibited the activation of NOD-like receptor thermal protein domain-associated protein 3 (NLRP3) inflammasome by decreasing the protein expression of NLRP3 and Caspase-1, and the gene expression of *Caspase*-*1*, *IL-1β* and *IL*-*18*. Meanwhile, both LPC and LPB observably activated autophagy under ST infection, as indicated by the up-regulated expression of LC3 and Beclin1 and the downregulated p62 level (*p* < 0.05). Finally, we found that LP postbiotics could trigger an AMP-activated protein kinase (AMPK) signaling pathway to induce autophagy. In summary, *Lactiplantibacillus plantarum*-derived postbiotics alleviated *Salmonella* infection via modulating bacterial pathogenicity, autophagy and NLRP3 inflammasome in mice. Our results confirmed the effectiveness of postbiotics agents in the control of *Salmonella* infection.

## 1. Introduction

Probiotics have been broadly used in animals for numerous benefits [1]. However, some probiotics might produce toxins and possess antimicrobial resistance genes, which can be a potential risk for animal health [2]. Additionally, the shortcomings of probiotics, such as their intolerance to high temperatures, low pH and long-time storage, have also limited their application [3]. Evidence has shown that the metabolites and bacterial components of probiotics remain highly physiologically active after treatment with high temperatures or gastrointestinal digestive juices [4,5]. These components are termed “postbiotics”, defined as “preparation of inanimate microorganisms and/or their components that confers a health benefit on the host” [6]. Accumulating research has indicated that postbiotics have similar or even better effects than their active bacteria [7]. Compared with live strains, postbiotics of *Saccharomyces boulardii* showed better efficacy in alleviating dextran sulfate sodium-induced colitis by modulating inflammation and intestinal microbiota in mice [8]. The killed lactic acid bacteria exhibited better immunomodulatory activities than live probiotic in mice spleen cells [9]. The heat-inactivated *Lactobacillus gasseri* increased immunomodulatory activity more than the live strains in macrophages [10].

*Salmonella* is one of the most important pathogens which has threats to food safety and animal and human health. It is able to cause intractable septicemia, anemia, bacteremia, meningitis and fatal dysentery [11]. At present, the main treatment method for *Salmonella* is using antibiotics. However, the abuse of antibiotics can lead to bacterial resistance and other side effects. Recent studies have reported the anti-*Salmonella* effect of postbiotics. For example, *Lactobacillus rhamnosus SQ*511 culture supernatant directly inhibited *Salmonella* growth [12]; *Lactobacillus* postbiotics improved growth performance and enhanced immunity of broiler chickens after *Salmonella* infection [13]. However, their mechanism of action remains to be investigated.

Activating the host inflammatory response is one of the main manifestations of *Salmonella* infection [14]. Inflammasomes play a complex and important influence in activating and releasing inflammatory factors. Among them, NOD-like receptor thermal protein domain-associated protein 3 (NLRP3) inflammasome has attracted widespread attention as a protein complex that can be activated by multiple stimuli and make a vital difference to the immune function regulation and inflammatory signaling [15]. Pathogen-associated molecular patterns (PAMPs) trigger a signaling cascade reaction that causes cytosolic pattern-recognition receptors (PRRs) like NLRs to form the “the inflammasome” complexed with multimeric protein [16]. *Salmonella* interaction with the pyrin structural domain triggers the recruitment of apoptosis-associated speck-like proteins containing a CARD (ASC) by NLRP3, which increases pro-Caspase-1 through CARD–CARD interactions and activates it to form Caspase-1 and ultimately, NLRP3 inflammatory vesicles. Then, activated Caspase-1 cleaves pro-IL-1β and pro-IL-18 into mature forms, ultimately leading to an inflammatory response [17]. Restraining NLPR3 inflammasome activation is therefore crucial to suppress *Salmonella*-mediated inflammation. 

Autophagy is closely related to the inhibition of inflammation. When NLRP3 is activated, ASCs are recognized by the adaptor protein p62, bound to autophagosome and eventually degraded. Peng et al. [18] confirmed the relation of autophagy and inflammasome. Therefore, triggering autophagy to inhibit NLRP3 inflammasome may be essential to control inflammatory responses during *Salmonella* infection. Numerous studies have shown that autophagy exerts a key role in the host resistance to *Salmonella* [19,20]. Postbiotics have been shown to have a modulatory effect on autophagy. *L. paracasei* culture supernatants stimulated autophagy of Caco-2 cells [21]; postbiotics of *L. fermentum* alleviated acetaminophen-induced hepatic intoxication through activating autophagy in HepG2 cells [22]. Therefore, the present research aimed to investigate the effectiveness of postbiotics from *Lactiplantibacillus plantarum* (formerly known as *Lactobacillus plantarum*) [23] on protecting mice against *Salmonella* infection and elucidate the underlying mechanisms modulating NLRP3 inflammasome and autophagy. 

## 2. Materials and Methods 

### 2.1. Bacteria Preparation

*Lactiplantibacillus plantarum* HJZW08 (CGMCC: No. 23777) was provided by Zhejiang Vegamax Biotechnology Co., Ltd. (Huzhou, China) and was cultivated in MRS broth at 37 °C for 24 h. The active bacteria (LPB_active_) were rinsed in PBS and transferred into new MRS broth to eventually reach 1 × 10^9^ cfu/mL, and the LPB_inactive_ were then prepared by heating at 100 °C for 15 min. LP metabolites (LP culture supernatant, LPC) were obtained by centrifugation and sterile filtration. NaOH-LPC was obtained by adjusting the pH of LPC to 6.5 with 0.1 mol/L NaOH. Catalase-LPC was obtained by removing H_2_O_2_ via incubation with catalase (1 mg/mL) at 25 °C for 1 h. Protease-LPC was obtained by removing bacteriocins with trypsin (200 mg/mL) and protease K (1 mg/mL) at 37 °C for 1 h. Heat-LPC was obtained at 100 °C for 15 min. *Salmonella* Typhimurium (ST) SL1344 was cultivated in LB broth overnight, then washed in PBS and diluted to 3 × 10^9^ cfu/mL.

### 2.2. Animal Experimental Design

The mice experiments were conducted according to the Guide for the Care and Use of Laboratory Animals, and all procedures were approved by the Animal Care and Use Committee at Zhejiang Agriculture and Forestry University (Approval number: ZAFUAC2022011). Male C57BL/6 mice of 5 weeks were purchased from SLAC Laboratory Animal Co., Ltd. (Shanghai, China). They were raised in a biosafety and comfortable environment. The environmental conditions were temperature at 21 ± 1 °C and humidity at 50~60%, with a 12 h light/dark cycle. They were supplied with the standard forage and filtered water and allowed to eat and drink freely. The eighty mice were divided into five groups (Control, ST, LPB_active_ + ST, LPB_inactive_ + ST and LPC + ST). After a seven-day acclimatization period, mice were severally given intragastric administration of 0.2 mL MRS broth, LPB_active_ (1 × 10^9^ cfu/mL), LPB_inactive_ (1 × 10^9^ cfu/mL) or LPC every day. Following a 15-day pretreatment period, the ST, LPB_active_ + ST, LPB_inactive_ + ST and LPC + ST groups were all gavaged by 8-gauge gavage needles with 0.1 mL of ST suspension (3 × 10^9^ cfu/mL) in PBS, while the control group received equal volume of PBS. The animals were then sacrificed by ether anesthesia three days later. All mice samples were instantly frozen in liquid nitrogen and stored at −80 °C.

### 2.3. Effects of LPC on ST Growth

#### 2.3.1. Turbidimetry Test

The *Salmonella* (2.5 × 10^8^ cfu/mL) was co-cultured with gentamicin (GM, 25 μg/mL) or LPC at concentrations from 1% to 9% at 37 °C. The OD_600_ absorbance of each group was measured at 0, 2, 4, 6, 8, 10 and 12 h, and the time-OD curve was drawn. 

#### 2.3.2. Agar Diffusion Assay

The plates were filled with LB solid medium with 5% agar as the bottom layer. Approximately 10^8^ cfu/mL of ST cells were then added into in the LB agar (1% *w*/*v*), and then, the mixture was transferred as the upper layer. Then, sterile oxford cups were placed on the solidified agar, and 250 μL MRS, 2% LPC and GM (25 μg/mL), respectively, were transferred into each well. The anti-*Salmonella* activity was estimated by measuring the diameter of the inhibition zone. NaOH-LPC, Catalase-LPC, Protease-LPC and Heat-LPC groups were added to detect the antibacterial components of LPC by the above method.

### 2.4. Biofilm Inhibition Assay

ST was co-cultured with 2% LPC or GM (25 μg/mL) in 24-well plates for 12 and 24 h. Then, planktonic ST cells were removed by sucking out the supernatant and washing three times with PBS gently. Then, plates were stained with 0.1% (*w*/*v*) crystal violet at 23–25 °C for 30 min. Then, every well was given 3–4 rinses using distilled water and dried at room temperature for 15 min. After incubating with 95% (*v*/*v*) ethanol for 30 min, the absorbance was determined by an automated microplate reader (SynergyH1, BioTeK, Winooski, VT, USA) at 545 nm [24]. The equation of biofilm inhibition is as follows:$\;Biofilminhibitionrate\left(\%\right)=\frac{{OD}_{Salmonella}-{OD}_{testgroup}}{{OD}_{Salmonella}}\times 100$.

OD*_Salmonella_*: contains only the OD value of *Salmonella*; OD _test group_: LPC+ *Salmonella* or GM+ *Salmonella* OD value.

### 2.5. Intestinal Morphological Analysis 

A small section of middle ileum was fixed with paraformaldehyde at 4%, buried in paraffin and sliced to 3 μm, and then stained using hematoxylin and eosin (H&E). Then, images were photographed through a Nikon microsystem (Nikon, Tokyo, Japan). The villus length and crypt depth were statistically calculated from 8 fields per slide and 8 slides in each group.

### 2.6. Inflammatory Cytokines Analysis

The blood samples were immediately collected after mice sacrifice and the serum was obtained by centrifuging at 4000× *g* for 10 min and stored at −80 °C for further study. The ileum tissues were homogenized with PBS, followed by a centrifugation (10,000× *g*, 4 °C) for 15 min to obtain the supernatant. The contents of IL-1β, TNF-α, IL-6, IL-4 and IL-10 were detected using ELISA commercial kits (Angle Gene ELISA kits, Nanjing, China).

### 2.7. Western Blot

Mice ileum samples were homogenized in NP-40 (Beyotime, Shanghai, China) with 1% PMSF and phosphatase inhibitors (Beyotime, Shanghai, China) on ice for half an hour. The BCA kit (Beyotime, Shanghai, China) was used to quantify protein concentration. After centrifuging and denaturating, 20 μg protein of each sample was added to the well and separated using 12% SDS polyacrylamide gels at 80 V for 30 min, then at 120 V for 1 h, and transferred onto polyvinylidene difluoride (PVDF) membranes of 0.22 μm or 0.45 μm pore size (Millipore, Carlsbad, CA, USA) at 260 mA for 1.5 h. Afterwards, the membranes were blocked by 5% (*w*/*v*) skimmed milk and cultivated with corresponding rabbit-derived primary antibody at 1:1000 overnight at 4 °C. Primary antibodies against LC3, p62, Beclin1, NLRP3, Caspase-1, Occludin, Claudin-1, AMPK, p-AMPK, TAK1, ULK1 and β-actin were purchased from Hua’an Technology (Hua’an, Hangzhou, China), and p-ULK1 was purchased from Cell Signaling Technology (Cell Signaling Technology, Danvers, MA, USA). After incubating with the secondary antibody HRP-labeled Goat Anti-Rabbit IgG (H+L) (Beyotime, Shanghai, China) at 1:1000 for 1 h, a Tanon 4600 series automatic chemiluminescence image analysis system (Tanon, Shanghai, China) was applied to detect the immunoreactive bands. Eventually, the ImageJ software v1.8.0. was used to determine the relative band density, and β-actin was used as the control. 

### 2.8. mRNA Relative Expression Analysis by Real-Time PCR Assay

The total RNA was extracted from the *Salmonella* and C57BL/6 mice ileum using RNAiso plus (Takara Bio Inc., Otsu, Japan), referring to the manufacturer’s instructions. Prime Script^TM^ RT reagent Kit with g DNA Eraser kit were used to reverse transcribe 1 μL RNA (1000 ng/μL) into cDNA. The reaction conditions were 42 °C for 2 min to remove the DNA, then 37 °C for 15 min and 85 °C for 5 s to synthesize cDNA (50 ng/μL). Real-time PCR was performed in a CFX real-time PCR system (Bio-Rad Laboratories, Hercules, CA, USA) using SYBR Premix Ex TaqII (Takara Bio Inc., Otsu, Japan) under the following conditions: 95 °C for 30 s for initial denaturation, followed by 40 cycles at 95 °C for 5 s and 60 °C for 30 s. β-actin and 16sRNA were regarded as reference genes. mRNA relative expression levels were calculated using the 2^−ΔΔCT^ method (ΔCT = CT_target gene_ − CT_reference gene_; ΔΔCT = ΔCT_treated sample_ − ΔCT_control sample_) [25]. The primers are as shown in Table 1.

### 2.9. Statistical Analysis

All values are presented as means ± standard error. GraphPad Prism 8.0 was used to draw figures. Differences were conducted by one-way ANOVA and Tukey’s test (SPSS Inc., Chicago, IL, USA). *p*-values of < 0.05 were deemed to be statistically significant. 

## 3. Results 

### 3.1. Effects of LPC on Salmonella Growth

As shown in Figure 1a, LPC significantly inhibited the growth of ST (*p* < 0.05) and the inhibition zone was 23 mm. Results in Figure 1b show that LPC at the concentrations of 2–9% inhibited the growth of *Salmonella* starting from co-cultured with 4 h, while 1% LPC showed no antibacterial effect. Moreover, when incubated with 7–9% LPC, the inhibitory effect was similar to that of GM. As displayed in Figure 1c, we further found that when LPC was neutralized by NaOH, the antibacterial ability was completely blocked, whereas when treated with protease, catalase or heat showed no obvious changes. These results manifested that the main antibacterial ingredient of LPC was organic acids.

### 3.2. Effects of LPC on Salmonella Pathogenicity

Figure 2a shows that 2% LPC notably restrained the relative expression levels of SPI-1 virulence genes *SopE*, *SopB*, *InvF*, *InvA*, *SipA*, *SipB* and *HilA* (*p* < 0.05). Compared to ST, the mRNA level of *SopD2* encoded by SPI-2 was markedly decreased by LPC treatment (*p* < 0.05). As shown in Figure 2b, 2% LPC significantly inhibited the expression of pilus assembly genes including *FilF*, *SefA*, *LpfA* and *FimF,* compared with ST (*p* < 0.05). Figure 2c indicates that 2% LPC significantly inhibited the expression of flagella genes *FlhD*, *FliC* and *FliD,* compared to ST (*p* < 0.05). As demonstrated in Figure 2d, the inhibition ratio of 2% LPC on ST biofilm formation reached 80% at 12 h and 24 h, and the inhibition effect was extremely notable (*p* < 0.05).

### 3.3. Effects of LP Postbiotics on Salmonella-Induced Intestinal Injury in Mice

Figure 3a indicates that all the LP pretreatments significantly decreased *Salmonella* colonization in the ileum (*p* < 0.05), and LPC exerted a better effect with a marked decrease than LPB_active_ (*p* < 0.05). As shown in Figure 3b, ST infection destroyed the intestinal villus construction and significantly reduced the length of the ileal villi and the villius/crypt ratio (*p* < 0.05). However, LP postbiotics and the active probiotic significantly reversed the trend by decreasing the crypt depth and increasing the villus/crypt ratio (*p* < 0.05). LPB_inactive_ significantly reduced crypt depth and increased the villus/crypt ratio compared to LPB_active_ and LPC (*p* < 0.05). Figure 3c shows that the expression of Occludin and Claudin-1 was significantly reduced after ST infection (*p* < 0.05), while pretreatments with LP postbiotics as well as LPB_active_ significantly reversed this trend (*p* < 0.05). The expression level of protein Occludin notably maintained a higher level in LPB_inactive_ + ST and LPC + ST groups than that of LPB_active_ (*p* < 0.05). 

### 3.4. Effects of LP Postbiotics on the Levels of Inflammatory Cytokines in Mice under Salmonella Challenge

As shown in Figure 4a, ST challenge significantly increased the serum pro-inflammatory factors, IL-1β, IL-6 and TNF-α (*p* < 0.05), and significantly decreased the levels of anti-inflammatory cytokines, IL-4 and IL-10 (*p* < 0.05). LP pretreatments significantly reversed this trend (*p* < 0.05). Interestingly, LPB_inactive_ was superior to LPB_active_ and LPC in inhibiting ST-induced inflammation, and LPC was superior to LPB_active_. Figure 4b demonstrates the results for ileal inflammatory factors, similar to the trend for inflammatory factors in serum. Both LP postbiotics and its live bacterial pretreatments significantly downregulated the contents of pro-inflammatory factors, IL-1β and IL-6, raised by ST infection (*p* < 0.05). LPB_inactive_ was more effective than LPB_active_ and LPC.

### 3.5. Effects of LP Postbiotics on NLRP3 Inflammasome in Mice under Salmonella Challenge 

As shown in Figure 5a, ST infection markedly improved the mRNA relative expression levels of inflammasome biomarkers including *Caspase-1*, *IL-1β* and *IL-18* compared with the control group (*p* < 0.05). Nevertheless, LP postbiotics and its live bacteria significantly downregulated them (*p* < 0.05). Furthermore, Figure 5b shows that LP postbiotics and the live bacteria acted through depressing the assembly of NLRP3 inflammasome and suppressing the inflammation caused by ST infection. LPC was superior to LPB_active_ and LPB_inactive_. LPB_inactive_ and LPC significantly inhibited Caspase-1 expression (*p* < 0.05) compared to ST, with a better inhibitory effect than that of LP live bacteria.

### 3.6. Effects of LP Postbiotics on Autophagy under Salmonella Challenge

The results in Figure 6a show that preliminary treatments of LP markedly increased the expression levels of Beclin1 and LC3-II, inhibited p62 expression compared to ST (*p* < 0.05) and activated cellular autophagy. In addition, LPB_inactive_ and LPC were significantly more effective than LPB_active_ on LC3-II (*p* < 0.05). The results suggest that LP postbiotics pretreatments, particularly LPC, activated autophagy to counteract ST and was more effective than its active bacteria. As shown in Figure 6b, LP postbiotics and its live bacteria pretreatments significantly enhanced the expression levels of proteins such as TAK1, p-ULK1 (Ser757) and p-AMPK compared to the ST group (*p* < 0.05), and LPB_inactive_ and LPC appeared to be stronger than LP active bacteria (Figure 6b). Thus, LP postbiotics promoted autophagy in mice through the AMPK/ULK1 signal pathway.

## 4. Discussion

Although numerous studies have shown the beneficial effects of probiotics, their potential disadvantages have gradually been recognized as research has deepened. Postbiotics are the inactivated bacteria and their metabolites of probiotics. Of these, the active components are mainly peptidoglycan and lipophosphate walls, and the metabolites are organic acids, hydrogen peroxide and bacteriocins [26,27]. The inhibitory effect of postbiotics on pathogenic bacteria has been proved. For example, postbiotics of *L. acidophilus* and *Enterococcus faecalis* observably reduced the quantity of *Clostridium perfringens* in the chicken digestive tract [28]. In this study, we found that postbiotics derived from *Lactiplantibacillus plantarum* significantly alleviated *Salmonella* infection by inhibiting bacterial pathogenicity and modulating autophagy and NLRP3 inflammasome in mice. 

Our results showed that LP metabolites markedly inhibited ST growth. The growth inhibitory effect of LPC on ST began at a concentration of 2% and intensified with increasing concentration. High-concentration LPC had even the same effect as GM, which provided evidence for the study of metabolites as an alternative to antibiotics. Similarly, it was reported that the postbiotics of *Lactiplantibacillus pentosus* SLC13 had a good bacteriostatic effect on the *Helicobacter pylori* in a dose-dependent manner [29]. Our study showed that the antibacterial substance produced by LPC was organic acids, which was in accordance with the outcomes of Russo et al. [30], and that the organic acids contained in *Lactobacillus plantarum* had a good suppressive effect on common foodborne pathogenic bacteria and spoilage bacteria. LPC was not sensitive to enzyme inhibitors (trypsin, proteinase K, catalase), indicating that the main antibacterial substances were not bacteriocin and hydrogen peroxide. Furthermore, we found that LPC had high thermal stability, which was identical to the results of Mekky et al. [31]. In summary, the inhibitory substance in LPC was an organic acid, but the specific substance that exerted the inhibitory effect needs to be further studied.

LP metabolites exerted a strong capacity in suppressing *Salmonella* pathogenicity. The pathogenicity of *Salmonella* is driven by virulence factors, flagella, pili and biofilm, which not only help the bacteria to invade cells and colonize, but also protect the bacteria from the external environment [32]. We found that LPC significantly decreased the mRNA expression of the virulence genes (*SopE, SopB, InvA, InvF, SipB, HilA, SipA and SopD2*). The virulence genes *SopB*, *SopE*, and *HilA* encoded by SPI-1 play key roles in *Salmonella* invasion of cells and cause inflammation. SPI-2 encodes *SopD2*, which contributes to its evasion of lysosomal degradation [33]. These virulence genes play a decisive part in invading host cells and the pathogenicity of *Salmonella*. Consistent with our results, it has been reported that *Lactococcus* suppressed *Salmonella* virulence genes’ expression levels [34]. Flagella is the motor organ of *Salmonella* which enables it to move forward in a fluctuating environment and is also an important virulence factor mediating bacterial attachment and invasion [35]. In the course of *Salmonella* infection, pili plays an important role in host recognition, colonization and biofilm formation, which is considered to be the main organelles mediating the interaction and adhesion between *Salmonella* and host intestinal epithelium [36]. In this study, LPC markedly repressed the gene expression involving in flagella (*FlhD, FliC, FliD*) and pili (*FilF, SefA, LpfA, FimF*) indicating its ability. Shi et al. [37] found that the culture supernatant of *Lactobacillus reuteri* S5 resulted in a significant reduction in the expression of *Salmonella* virulence, motility and adhesion genes and had an inhibitory effect on biofilm formation. We further found that LPC could inhibit the formation of ST biofilm. After colonizing the host intestine through adhesion and invasion, *Salmonella* further forms a dense biofilm to enhance its pathogenicity [38]. Bacteria in biofilms are generally well protected from environmental stress, antibiotics and the host immune system, making them extremely difficult to eradicate [39]. The bacterial resistance to antibiotics is 1000 times higher than that of planktic bacteria, leading to long-term chronic infection [40]. Previous studies have also shown inhibitory effects of probiotic culture supernatants on *Salmonella* biofilms [41,42]. Therefore, the results showed that LPC reduced the gene expression of ST virulence, pili and flagella, and suppressed biofilm formation to inhibit the pathogenicity of *Salmonella*.

The mice experiment revealed that LP postbiotics showed a similar or even superior effect than the active probiotic in protecting against *Salmonella* infection. LP postbiotics, particularly the metabolites, significantly decreased ST colonization in mice ileum. After invading the host, *Salmonella* first invades the intestinal epithelial cells of the body, reaches the lamina propria of the intestinal wall and multiplies in them, causing inflammation at the same time [43]. With the increase in the number of bacteria, damage and inflammation will be aggravated and transferred to other tissues. In line with our results, postbiotics of *Bacillus subtilis* and *Bacillus licheniformis* have been found to decrease the number of *Salmonella* in the gut and thus relieve damage [44]. *Salmonella* infection can cause intestinal mucosal and villi damage. We found that ST infection significantly decreased the villus length and increased the crypt depth, while pretreatments with LP postbiotics and the active bacteria could reverse this trend. The increase in villi height can enlarge the absorption area of small intestine, which is conducive to absorbing more nutrients [45]. Crypt depth reflects the cell formation rate, and the shallower crypt indicates increased cell maturation rate and secretion function [46]. Thus, higher villus height to crypt depth ratio indicates an increased absorptive capacity and cell formation rate to maintain the intestinal health. Previous studies have shown that probiotics play a crucial part in mitigating intestinal tissue damage [47,48], while there has been less research on postbiotics. Intestinal epithelial cells are made up of simple columnar intestinal epithelial cells whose gaps have many tight junction proteins (Occludin, Claudin, et al.), which are closely related to permeability [49]. Previous studies have shown that *Salmonella* infection led to intestinal damage, increased permeability and downregulated the expression of Occludin and Claudin in mice [50]. Consistently, we also found that ST infection dramatically reduced the expression of Occludin and Claudin-1, whereas LP postbiotics as well as the live probiotic could prevent the decrease, indicating their ability to enhance mucosal barrier. In summary, LP postbiotics showed a similar effect as the live probiotic in reducing ST colonization, alleviating villi damages, enhancing tight junction expression and ultimately, preventing *Salmonella*-induced gut injuries.

We found that LP postbiotics inhibited NLRP3 inflammasome to alleviate *Salmonella*-induced inflammation. *Salmonella* infection can activate inflammatory pathways by binding to TLRs on the cell surface to produce pro-inflammatory cytokines that cause an inflammatory response [51]. Our results showed that LP postbiotics pretreatments significantly decreased the levels of pro-inflammatory cytokines, IL-1β, TNF-α and IL-6, and increased anti-inflammatory cytokines, IL-4 and IL-10, suggesting its role in anti-inflammation. Previous studies have shown that the anti-inflammatory effect of postbiotics has been confirmed. For example, metabolites of *L. acidophilus* and *L. casei* could resist inflammation by lowering TNF-α cytokine level and increasing IL-10 level [52]. In addition, we found that the inactive bacteria showed stronger anti-inflammatory abilities than the live bacteria and the metabolites. Combined with previous studies on resisting inflammatory nature of lipoteichoic acid [53], we speculated that the anti-inflammatory component in postbiotics might be lipoteichoic acid. NLRP3 inflammasomes are typical inflammasomes that can be activated by *Salmonella*, and the expression of genes and proteins of its biomarkers could be inhibited by LP postbiotics. Consistent with our study, Huang and Lu et al. [54,55] found that postbiotics (active peptides and inactivated organisms) could inhibit the activation of inflammatory vesicles and attenuate *Salmonella*-induced intestinal inflammation. Therefore, our results suggest that LP, especially postbiotics, inhibited *Salmonella*-induced inflammatory responses and the activation of inflammasomes.

Our study demonstrates that LP postbiotics activated autophagy to protect against *Salmonella* infection. Autophagy is a significant innate immune recognition mechanism against pathogens. The autophagosome can recognize intracellular pathogens as a foreign entity, isolate them and then transport them to the lysosome for degradation and removal, which is known as xenophagy [56]. During *Salmonella* infection, autophagy protects cells from bacterial invasion [57]. In this work, we found that LP postbiotics significantly prevented ST-disrupted autophagy, as evidenced by the increased levels of LC3 and Beclin1 and the decreased p62. Similar with our results, recent studies have confirmed the activation of autophagy by probiotics upon pathogen infection. For example, *Lactobacillus yohimbe* prevented *Salmonella*-induced intestinal damage by regulating autophagy [58]; Exopolysaccharides from *Bifidobacterium animalis* ameliorated *Escherichia coli*-induced IPEC-J2 cell damage via inhibiting apoptosis and restoring autophagy [59]. We further found that AMPK/ULK1 signaling played a pivotal role in LP postbiotics-induced autophagy. AMPK/ULK1 and its upstream molecule TGF-β-activated kinase 1 (TAK1) [60] have a vital impact on cell survival, apoptosis, as well as inflammatory responses, and it is also a classic signaling pathway to activate autophagy. When cells are hungry or stimulated, activated AMPK can directly promote the phosphorylation of ULK1-related sites and ultimately initiate autophagy [61]. We found that LP postbiotics markedly upregulated the phosphorylation of AMPK and ULK1 and the expression of TAK1, suggesting their capacity of triggering autophagy. In line with our results, a growing number of studies have also revealed the ability of *Lactobacillus plantarum* to induce the AMPK signaling pathway [62]. Furthermore, it was reported that activation of AMPK/ULK1 was an effective way to defend against *Salmonella*. For instance, Zhuang et al. [63] demonstrated that autophagy could be induced by activating AMPK and ULK1 to restrict intracellular *S. Typhimurium* growth in vitro. Therefore, our study indicates that TAK1/AMPK/ULK1 plays a key role in LP postbiotic-activated autophagy to protect against *Salmonella* infection. 

## 5. Conclusions

In conclusion, LP postbiotics suppressed *Salmonella* infection via inhibiting bacterial pathogenicity and modulating autophagy and NLRP3 inflammasome in mice. Our findings confirm the antimicrobial activity of postbiotics and provide a strategy to prevent *Salmonella* infection in the livestock production. However, which components of LP postbiotics (the specific organic acids or bacterial components) exert the key effects warrants further investigation.

## Figures and Tables

**Figure 1 animals-13-03215-f001:**
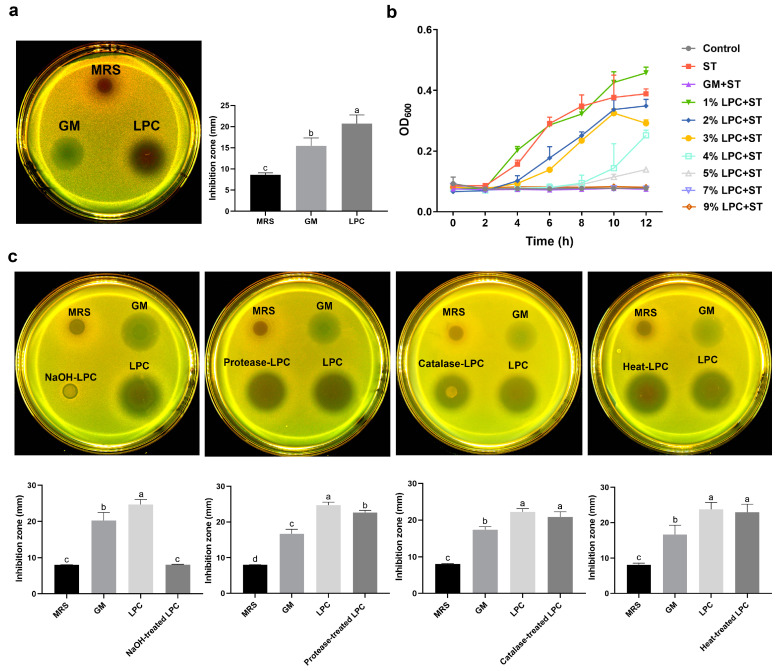
Effects of LPC on *Salmonella* Growth. (**a**) The inhibition effect of LPC on ST was detected via agar-well diffusion method; (**b**) inhibitory effect of different concentrations of LPC on ST; (**c**) the ST inhibitory components in LPC were determined via the method described in (**a**). Data were analyzed by one-way ANOVA and Tukey’s test (*n* = 6). Different lowercase letters indicate statistical significance with *p* < 0.05.

**Figure 2 animals-13-03215-f002:**
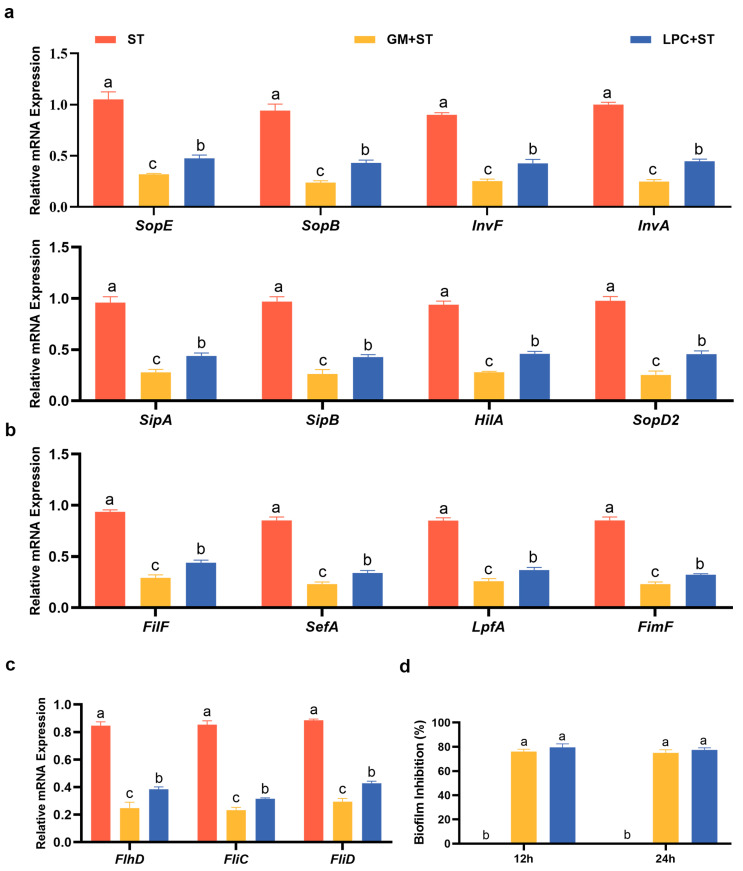
Effects of LPC on *Salmonella* Pathogenicity. (**a**–**c**) The relative mRNA expression levels of the virulence, pilli and flagellar related genes of ST; (**d**) the inhibition rates of ST biofilm. The concentration of LPC was 2%. Data analysis was conducted by one-way ANOVA and Tukey’s test (*n* = 3). Different lowercase letters indicate statistical significance with *p* < 0.05.

**Figure 3 animals-13-03215-f003:**
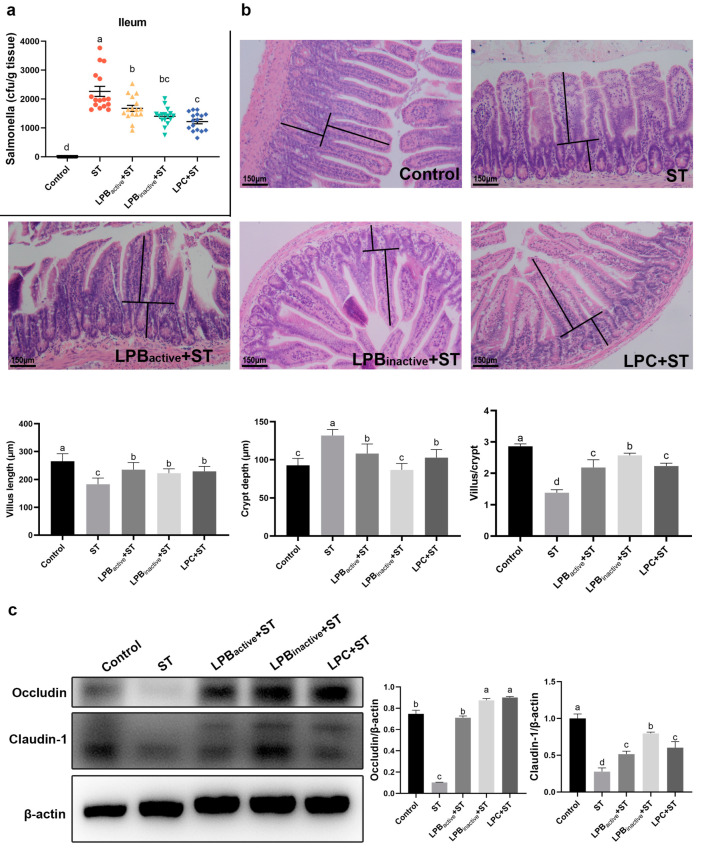
Effects of LP Postbiotics on *Salmonella*-Induced Intestinal Injury in Mice. (**a**) The amounts of ST in the ileum of mice; (**b**) effect of LPC on ileum villi, crypt and villus/crypt ratio in mice infected with ST; (**c**) effect of LPC on the tight junction proteins Occludin and Claudin-1 expression in mouse ileum. Data were carried out by one-way ANOVA and Tukey’s test (*n* = 3). Different lowercase letters indicate statistical significance with *p* < 0.05 (original western blot figures in Appendix A).

**Figure 4 animals-13-03215-f004:**
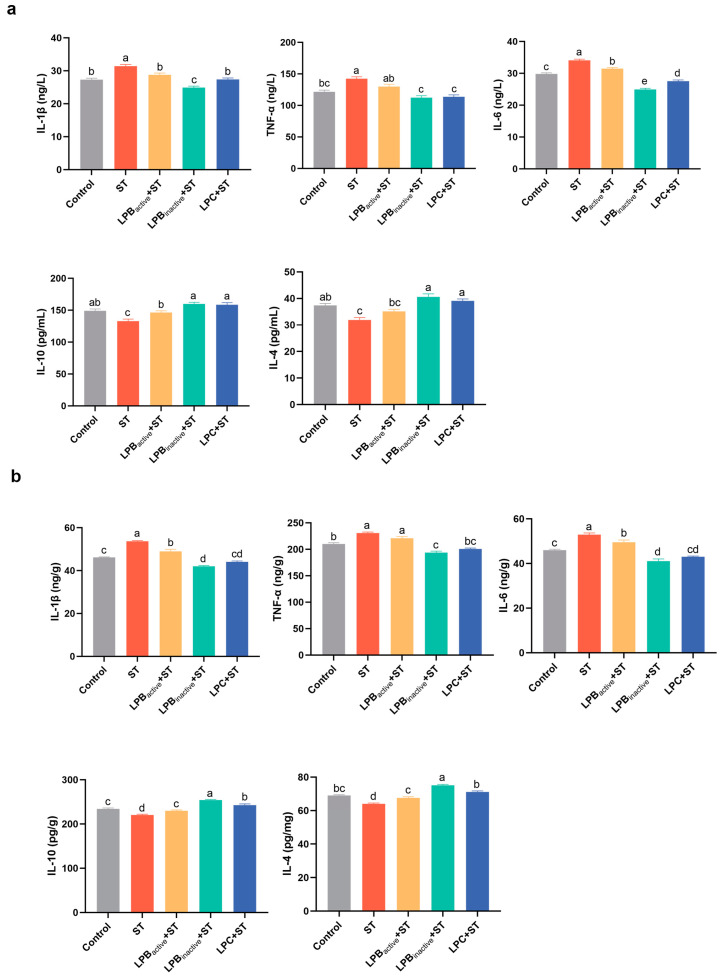
Effects of LP Postbiotics on the Levels of Inflammatory Cytokines in Mice under *Salmonella* Challenge. (**a**) The contents of inflammatory factors in mouse serum; (**b**) the levels of inflammatory cytokines in mouse ileum. Data analysis was conducted by one-way ANOVA and Tukey’s test (*n* = 6). Different lowercase letters indicate statistical significance with *p* < 0.05.

**Figure 5 animals-13-03215-f005:**
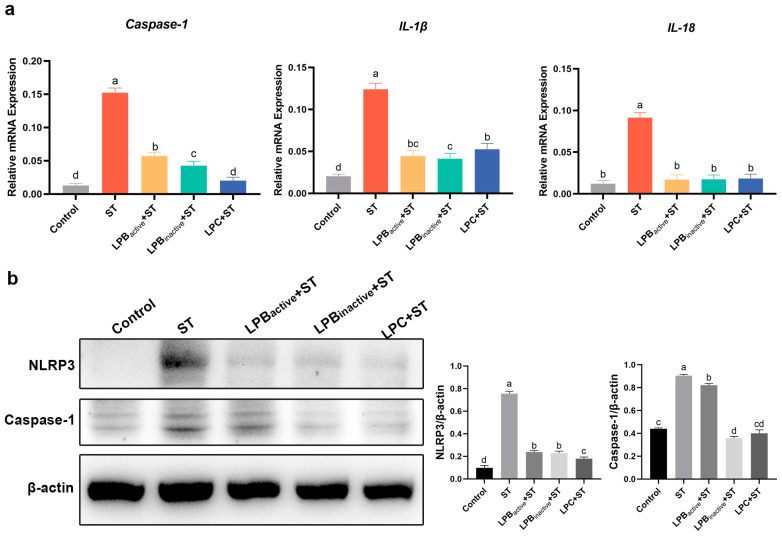
Effects of LP Postbiotics on NLRP3 Inflammasome in Mice under *Salmonella* Challenge. (**a**) The expression of inflammasome genes in mouse ileum; (**b**) the protein expression levels of NLRP3 and Caspase-1. Data were analyzed using one-way ANOVA and Tukey’s test (*n* = 3). Different lowercase letters indicate statistical significance with *p* < 0.05 (original western blot figures in Appendix A).

**Figure 6 animals-13-03215-f006:**
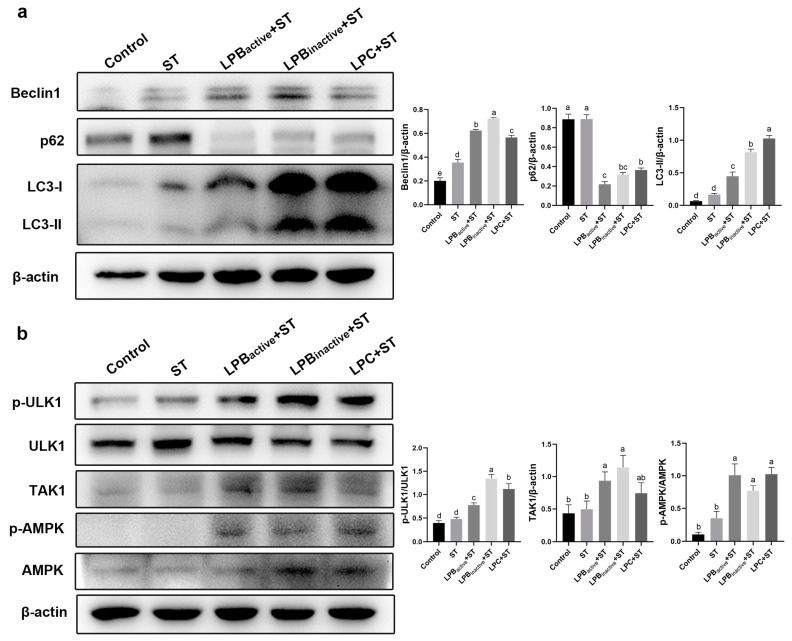
Effects of LP Postbiotics on Autophagy under *Salmonella* Challenge. (**a**) The protein expression levels of Beclin1, p62, LC3 and β-actin; (**b**) the protein expression of TAK1, p-ULK1, ULK1, p-AMPK, AMPK and β-actin. Data analysis was carried out by one-way ANOVA and Tukey’s test (*n* = 3). Different lowercase letters indicate statistical significance with *p* < 0.05 (original western blot figures in Appendix A).

**Table 1 animals-13-03215-t001:** Primer sequences used in real-time quantitative PCR ^1^.

Gene Names	Forward Primer (5′-3′)	Reverse Primer (5′-3′)
*InvA*	GCTTGGCTATGTGTTGCGGAAC	CGTGGCATGTCTGAGCACTTCT
*InvF*	TTTGCTGAGTCCTGAGTTTCGC	TCATCGTGTTGCCGCTGGTT
*SopE*	GCCAGACCCGTGAAGCTATACT	TCGCTGCTTCGCCAATTTCCT
*SopB*	GATGCCCGTTATGCGTGAGTGT	TCAGCAGCAGGATGGCTTACCT
*SipB*	GCTGATTGGCAAGGCGATTACC	CGACAACTGCGACCACCACAAT
*HilA*	AATCGTCCGGTCGTAGTGGTGT	TGCGGCAGTTCTTCGTAATGGT
*SipA*	CAACGCCACCAGTGATTCTCCT	GCTTCGCTTCCGCTTTCTTTGT
*SopD2*	AGCCCGTTTGATGAGTCCTG	ACCTCCAGCACCTCTTGTTT
*FliF*	GAAGCCATTCTGTCGCCTAT	TGTAGTGCTCTTCCGTCTGC
*LpfA*	TTTGCTCTGTCTGCTCTCGC	CTGACCCAGCACAACTTCCT
*SefA*	CTGTCCCGTTCGTTGATGGA	CTGCTGGCAGGGTCGATTTA
*FimF*	CCATTGCCGTATCAGCAAGC	ACAAAACAGCTTCACGTCGC
*FlhD*	CGCCTCGGTATCAACGAAGA	GCGCGAATCCTGAGTCAAAC
*FliC*	TCTGTCCTCTGGTCTGCGTA	TCATTCAGCGCACCTTCAGT
*FliD*	GCCAATTACCAAACAGCAGAG	GACGCCACGGTAGACTTAAATA
*16sRNA*	CGATGTCTACTTGGAGGTTGTG	CTCTGGAAAGTTCTGTGGATGTC
*Caspase-1*	GCTCCAACCCTCGGAGAAAG	AACCTTGGGCTTGTCTT
*IL-1β*	GCCACCTTTTGACAGTGATG	GATGTGCTGCTGCGAGATTT
*IL-18*	ACCTGAAGAAAATGGAGACCTGG	GGGGTTCACTGGCACTTTGA
*β-actin*	GATGTGCTGCTGCGAGATTT	AGGAAGGCTGGAAGAGTGC

^1^*InvA*, *Salmonella* invasion protein A; *InvF*, *Salmonella* invasion protein F; *SopE*, *Salmonella* pathogenicity island encoded protein E; *SopB*, *Salmonella* pathogenicity island encoded protein B; *SipB*, *Salmonella* cell invasion protein B; *HilA*, *Salmonella* invasion genes transcription activator A; *SipA*, *Salmonella* cell invasion protein A; *SopD2*, *Salmonella* pathogenicity island encoded protein D2; *FliF*, *Salmonella* flagellar biosynthesis protein F; *LpfA*, *Salmonella* long polar fimbria A; *SefA*, *Salmonella* fimbrial operon gene A; *FimF*, *Salmonella* putative fimbrial protein F; *FlhD*, *Salmonella* flagellar transcriptional regulator D; *FliC*, *Salmonella* flagellar filament capping protein C; *FliD*, *Salmonella* flagellar filament capping protein D; *16sRNA*, *Salmonella* 16S ribosomal RNA; *Caspase-1*, cysteinyl aspartate specific proteinase-1; *IL-1β*, interleukin-1 beta; *IL-18*, interleukin-18; *β-actin*, cytoskeletal actin. All primers were designed in this study.

## Data Availability

All data generated or analyzed during this study are included in this published article.

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
