# Peer review of "Lactiplantibacillus plantarum Postbiotics Suppress Salmonella Infection via Modulating Bacterial Pathogenicity, Autophagy and Inflammasome in Mice"

_animals, 2023, doi:10.3390/ani13203215_

Round 1

Reviewer 1 Report

The study of the products of cultivation of Lactobacillus plantarum is of significant practical interest throughout the world. The authors studied a topic that is also of considerable scientific interest. The interaction of the pathogenic bacterium Salmonella at the genetic level with the waste products of Lactobacillus plantarum remains unexplored. 

Manuscript flaws.

1. Keywords “Postbiotic; Lactobacillus plantarum; Salmonella infection" repeat the words of the title. It is better to write other 4-5 phrases instead of them, not used in the title of the article and abstract.

2. Line 110: Local ethics committee approval required. Be sure to briefly describe what procedures or conditions of detention could be inhumane to animals and how the authors minimized traumatic situations for animals (3-4 sentences).

3. Lines 198: Invalid small image. The picture needs to be changed so that the information on the lower histograms is at least 4-5 times larger. Photos of Petri Czechs are best removed. In the last century they were important, but in this century there are enough histograms with correct methods for comparing samples.

4. Lines 182, 203, 224, 245 et seq.: statistical methods are incorrectly described. Have you applied a correction for multiple comparison of samples (eg Bonferroni correction)? Without its use, the authors keep silent in 10-30% of cases, false-positive differences between the samples. When applying the Student's t-test, the authors do not indicate whether the estimation of the normality of the sample distribution was carried out. Of the 50-100 samples with a 6-fold repetition of the experiment, a third of the samples will have an abnormal distribution. What did the authors do if they found a significant asymmetry or kurtosis in the sample?

5. Line 197: If the standard error of the mean is correctly displayed on the bars, then in two cases I have questions comparing bars, the differences between which are less than 0.05. Were the Bonferroni amendments taken into account? The same questions arise for other figures.

6. Line 220: It is not necessary to repeat the same name of the y-axis and the same legend many times. It is necessary to name each of the coordinate systems with the letters a, b, c, ....

7. If there are more than 4 columns (as in Figure 3 and subsequent figures), then it is advisable to carry out Tukey's test and use the same letters (as done on line 201) to designate columns that do not differ significantly, and with different letters - those columns that are different.

8. Abbreviations of the titles of the journals are carried out very carelessly, many data of publications are missing (lines 413, 415, 426, 431, 478 and many others). This greatly spoils the impression of a generally professionally performed study.

9. Line 353: In the second case, the generic name should be abbreviated.

10. Line 297: The discussion should be structured into subsections according to the subsections of the results. Each subsection of the results needs a separate, complete discussion, compared with existing publications.

11. Lines 381-384: The conclusion is unintelligible. It is necessary to look at the manuscript from a greater distance, to compare the results in a broader context, to outline possible further prospective studies.

Author Response

Reviewer 1

The study of the products of cultivation of Lactobacillus plantarum is of significant practical interest throughout the world. The authors studied a topic that is also of considerable scientific interest. The interaction of the pathogenic bacterium Salmonella at the genetic level with the waste products of Lactobacillus plantarum remains unexplored. 

Manuscript flaws.

  1. Keywords “Postbiotic; Lactobacillus plantarum; Salmonella infection" repeat the words of the title. It is better to write other 4-5 phrases instead of them, not used in the title of the article and abstract.

Many thanks to your good suggestions. We have replaced the keywords as “Probiotics; Inactivated bacteria; Metabolites; Salmonella Typhimurium; Autophagy; Inflammatory response”. Please check Lines 52-53.

  1. Line 110: Local ethics committee approval required. Be sure to briefly describe what procedures or conditions of detention could be inhumane to animals and how the authors minimized traumatic situations for animals (3-4 sentences).

Thank you for your good advice. We have added the local ethics committee approval number and described the conditions of detention. During the experiment, we have tried to minimized traumatic situations for animals and safeguard animal welfare. Please check Lines 123-130.

  1. Lines 198: Invalid small image. The picture needs to be changed so that the information on the lower histograms is at least 4-5 times larger. Photos of Petri Czechs are best removed. In the last century they were important, but in this century there are enough histograms with correct methods for comparing samples.

Thank you very much for your good advice. We have redrawn Figure 1 and tried to present clearer images. As for the photos of Petri Czechs, we might not want to remove them, because we think they can more directly show the inhibition zone of different treatments. Thank you very much for your understanding.

  1. Lines 182, 203, 224, 245 et seq.: statistical methods are incorrectly described. Have you applied a correction for multiple comparison of samples (eg Bonferroni correction)? Without its use, the authors keep silent in 10-30% of cases, false-positive differences between the samples. When applying the Student's t-test, the authors do not indicate whether the estimation of the normality of the sample distribution was carried out. Of the 50-100 samples with a 6-fold repetition of the experiment, a third of the samples will have an abnormal distribution. What did the authors do if they found a significant asymmetry or kurtosis in the sample?

Thank you very much for your careful examination. We are sorry for our mistakes. After our further careful checking, we found that we had written the wrong statistical method. In fact, we used one-way ANOVA and Tukey’s test to analyze the data.

  1. Line 197: If the standard error of the mean is correctly displayed on the bars, then in two cases I have questions comparing bars, the differences between which are less than 0.05. Were the Bonferroni amendments taken into account? The same questions arise for other figures.

Thanks for your question. We have revised all statistical method to one-way ANOVA and Tukey’s test in the revised manuscript. We took Tukey amendments into account when the differences were less than 0.05.

  1. Line 220: It is not necessary to repeat the same name of the y-axis and the same legend many times. It is necessary to name each of the coordinate systems with the letters a, b, c, ....

Thank you very much for your good advice. We have revised Figure 2 and the legends in accordance with your suggestions.

  1. If there are more than 4 columns (as in Figure 3 and subsequent figures), then it is advisable to carry out Tukey's test and use the same letters (as done on line 201) to designate columns that do not differ significantly, and with different letters - those columns that are different.

Many thanks to your good suggestion. We are sorry for our carelessness. After carefully examined, actually, all the data were analyzed by one-way ANOVA and Tukey’s test. We have corrected the statistical methods and re-organized all the pictures using letters to present the differences according to your advice.

  1. Abbreviations of the titles of the journals are carried out very carelessly, many data of publications are missing (lines 413, 415, 426, 431, 478 and many others). This greatly spoils the impression of a generally professionally performed study.

Thanks for pointing out our mistakes. We have carefully gone through all the references and corrected the journal abbreviations.

  1. Line 353: In the second case, the generic name should be abbreviated.

Thanks a lot for your reminder. All the generic names in the manuscript have been abbreviated.

  1. Line 297: The discussion should be structured into subsections according to the subsections of the results. Each subsection of the results needs a separate, complete discussion, compared with existing publications.

Thank you very much. Followed your suggestions, we have revised the Discussion section and tried to discuss the results fully. Please check the revised manuscript.

  1. Lines 381-384: The conclusion is unintelligible. It is necessary to look at the manuscript from a greater distance, to compare the results in a broader context, to outline possible further prospective studies.

Many thanks for your insightful advice. We have modified this section as “In conclusion, LP postbiotics suppressed Salmonella infection via inhibiting bacterial pathogenicity, and modulating autophagy and NLRP3 inflammasome in mice. Our findings confirmed the antimicrobial activity of postbiotics and provide a strategy to prevent Salmonella infection in the livestock production. However, which components of LP postbiotics (the specific organic acids or bacterial components) exert the key effects, warrants further investigation”.

Reviewer 2 Report

Brief Summary – The authors aimed to study the effects of Postbiotics from Lactobacillus plantarum cultures towards alleviating Salmonella infection un mice. They demonstrated that the inhibitory activity was not due to the live bacteria (heat treatment did not affect the activity), and the activity was not due to small peptides (bacteriocins) or Hydrogen peroxide produced by L. plantarum (L.P.). However, they found that the activity was due to “the organic acids” produced by LP. The authors went on to study the effects of LPC (LP culture supernatant) On the expression of Salmonella Pathogenicity Island-I (SPI-1)/inflammatory cytokines; as well as the effects of LPB-active, LPB-inactive, and LPC on survival of Salmonella-Challenged Mice and intestinal injuries.

 Weakness, and missing controls - The authors did not follow on an important observation they found - the activity was due to “the organic acids” produced by L.P.

Specific Comments:

1-    Citations are not accurate! For example References #2 and #3 - The claim that UN-FAO lately indicated that probiotics might possess a few ….[2,3] etc. 

Is not true! Please, check every citation for accuracy.

2-    Results – Figure 1a Vs Fig. 1b- The growth Kinetics are completely different for the ST in LB media. Fig 1a (Red line) is exponential (as it should be), while Fig 1b (red line) is linear growth – Why? 

3-    Figure 1d – NaOH completely neutralized the effect of LPC. The conclusion that Organic acids are the cause of the observed inhibition is correct. Identity of the organic acids!

4-    Figures 2 through Fig 6. Should include studies using NaOH treated LPC – in order to show if the observed effects are due to: the low pH,  a specific organic Acid, or else!!

5-    Discussion needs to be rewritten in light of the new data that will include NaOH – treated LPC.and/or a specific organic acid.

Can be improved.

Reviewer 3 Report

The article is very interesting since it proposes an innovative use of postbiotics from L. plantarum as alternative to antibiotics against Salmonella species. I really appreciated the idea, the amount of work and its presentation. I have only one major concern regarding the statistical analysis that probably should be revised or a more complete model should be considered, since the data are very fascinating and need to be properly valorized, and I am not sure that Student t-test was applicable in many cases (since you had 3 or more groups for most of the set of data).

I have some minor concerns regarding the introduction, and I think that more details should be provided to materials and methods in order to ensure the repeatability of results.

In general, correct the nomenclature of L. plantarum (and other bacteria that were previously categorized as Lactobacillus) according to the new guidelines that were recently introduced (Zeng et al., 2020; https://doi.org/10.1099/ijsem.0.004107).

In addition, I have some specific comments as follows below. 

Simple summary: This section should be expanded providing to readers more information about the experiment that was conducted, since only some results have been presented, try to describe your study with simple and not technical language.

Line 18: I think that NLRP3 inflammasome is a too technical definition that could result difficult for general audience that should read the simple summary. Please explain postbiotics in the simple summary with a brief definition, remembering that simple summary is not meant for the scientific sector.

Line 23: Are you sure that S. typhimurium is correctly stated in this sentence?

Line 24: Prior to presenting results, it would be useful to briefly describe materials and methods.

Line 27: When listing statistically significant differences the p-value should be added, check it for the entire abstract section.

Line 29: Try to be more precise, since “better effect” is too generic.

Line 32: The acronyms should be provided with the extended form the first time that they appear within the text. Interleukins are probably well known and they may be stated as IL, but NLRP3 could be difficult for readers with a different background. Please provide the in extenso form for NLRP3.

Line 57: Which strain of probiotic? Was it L. plantarum?

Line 61: Mortality rate?

Line 67: Enhanced.

Line 68: “their” mechanism of action.

Line 71: Rephrase the sentence listing the definition of NLRP3 before the acronym and put NLRP3 between brackets.

Lines 75-76: Use the correct style prior defining the acronym and put the abbreviation between brackets as you did for PAMPs. Check it for all abbreviations used in the text.

Line 86: Put the reference after the last name of author (Ge et al. [22]).

Line 90: Correct the typo.

Line 91: Correct the typo.

Line 100: Did you supplement MRS with glucose or other sugar source?

Line 102: How did you measure 1×109 CFU/mL.

Line 103: LP metabolites were obtained from both cultures or only from the inactivated L. plantarum?

Line 108: Please correct the parenthesis.

Line 111: Please provide the authorization number by the ethical committee.

Line 113: Please clarify what is mean for “standard forage”. Can you provide more information about the environmental conditions? Temperature, humidity, light?

Line 115: Please define groups, clearly explaining the treatments.

Line 116: Please describe the gavage conditions regarding the needle size.

Line 117: How did you select the dosage and time of administration of LBPactive and LPBinactive.

Line 120: Why not sterile LB?

Line 121: Which method was used for the sacrifice?

Line 125: How much volume was used for the following culture? Did you remove aliquots for checking OD at different times, or this test was conducted in microplate?

Line 138: How did you remove the planktonic cells?

Line 139: Can you provide a reference for this biofilm inhibition method?

Line 149: Please provide the thickness of slices.

Line 151: How many fields per slide were evaluated for villus length and crypt depth.

Lube 153: Did you collect blood immediately after sacrifice?

Line 156: “commercial kits”.

Line 160: Which buffer did you use for homogenization?

Line 161: Please provide the protocol information for electrophoresis run and protein transfer on PVDF membrane.

Line 164: “skimmed milk”. Which was the concentration of the blocking solution? Which was the concentration of primary antibody? Provide more information regarding the species and concentration. Which concentration was added to SDS gel? Did you quantify the extracted proteins with Bradford method?

Line 168: Which species, which concentration and how much time for the incubation of secondary antibody? Did you mean HRP-conjugated antibody?

Line 169: “applied”. Which equipment was used for acquiring images?

Line 170: Which control was used for comparing the relative band density?

Line 172: The total RNA was “extracted”.

Line 174: Please provide the reverse transcription and PCR conditions (time, temperatures, concentrations, for each reagent and step).

Line 175: Which formula was used for calculating 2−ΔΔCT with two reference genes. Did each plate include a standard sample used as reference?

Table 1: Each primer should be defined for the gene in the table footnotes. Do you have a reference for the following primers or they were designed for this experimental design?

Line 179: I am bit confused regarding the statistics. Did you use GraphPad Prism 8.0 or SPSS 21.0. In addition, which set of data was analysed by student t-test and ANOVA. According to the experimental design, did you consider the interaction between time and treatment and repeated measurements (e.g. turbidity test)? Which test was selected for post-hoc multiple comparisons?

Line 185: Chapter titles should reflect the materials and methods section; this statement is the relative finding of the turbidity analysis. Revise the titles used for the results chapter.

Line 186: Did you consider the quadratic effect for the increasing concentrations of LPC?

Line 188: Please remove not statistically significant p-values.

Figure 1a: It is not clear which are the statistically significant differences? How do you explain that 1% LPC+ST was higher than ST after 12 hours?

Figure 1b: Why did you consider only 2% LPC+ST? Can you provide the error range?

Figure 1d: I suggest checking the first figure from the left, since GM is very similar to LCP in other plates. Did you check the sensitivity of Salmonella to GM?

Line 203: Student t-test can compare only two groups; in each plate you have 4 samples. How many replicates of each plate did you prepare?

Line 204: Please delete (P > 0.05) since it is already defined as not significant.

Line 206-210: This paragraph should be moved to the discussion section.

Lines 212-213: Significantly decreased the expression level compared to which group/s? Try to be more precise.

Figure 2: Why did you evaluate only the effect of 2% LPC?

Line 224: I have some doubts regarding the student t-test used for these comparisons. Can you check the statistical method for these data?

Line 231: Revise the English language.

Line 232: Provide the p-value.

Line 237: Significantly inhibited stays for “increased”?

Line 239-240: This sentence suits better in the discussion section.

Line 245: Revise the statistical method.

Line 255: Downregulated.

Line 256: LPBinactive was more effective than?

Figure 5a: Are you sure that LPBactive, LPBinactive and LPC+ST are significantly different to each other’s?

Line 271-273: This sentence would be more appropriate for the discussion.

Line 350: This sentence would benefit from a reference.

Line 358: Are some studies that reported the presence of phosphatidic acid in L. plantarum?

Line 365: Can you provide more details regarding the autophagy phenomena?

Line 383: What is meant for “fresh method”?

Line 384: I suggest removing poultry since livestock already includes the avian species.

The English language is appropriate, there are only a few sentences that need to be revised in order to use a more appropriate terminology, as detailed within my report. 

Round 2

Reviewer 1 Report

The article can be recommended for publication.